# Impacts of Phase Noise on the Anti-Jamming Performance of Power Inversion Algorithm

**DOI:** 10.3390/s22062362

**Published:** 2022-03-18

**Authors:** Minglei Zhou, Qing Wang, Fangmin He, Jin Meng

**Affiliations:** National Key Laboratory of Science and Technology on Vessel Integrated Power System, Naval University of Engineering, Wuhan 430030, China; ytzhouzhou@163.com (M.Z.); hefangminemc@126.com (F.H.); mengjinemc@163.com (J.M.)

**Keywords:** adaptive beamforming, power inversion algorithm, anti-jamming, down-conversion, phase noise

## Abstract

Power inversion (PI) is a known adaptive beamforming algorithm that is widely used in wireless communication systems for anti-jamming purposes. The PI algorithm is typically implemented in a digital domain, which requires the radio-frequency signals to be down-converted into base-band signals, and then sampled by ADCs. In practice, the down-conversion circuit will introduce phase noises into the base-band signals, which may degrade the performance of the algorithm. At present, the impacts of phase noise on the PI algorithm have not been studied, according to the open literature, which is, however, important for practical design. Therefore, in this paper, we present a theoretical analysis on the impacts, provide a new mathematical model of the PI algorithm, and offer a closed-form formula of the interference cancellation ratio (ICR) to quantify the relations between the algorithm performance and the phase noise level, as well as the number of auxiliary antennas. We find that the ICR in decibel decreases logarithmically linearly with the phase noise variance. In addition, the ICR improves with an increasing number of auxiliary antennas, but the increment is upper-bounded. The above findings are verified with both simulated and measured phase noise data.

## 1. Introduction

Adaptive beamforming [1,2,3] is widely used for anti-jamming in wireless communications, which suppresses interferences by forming array beam nulls in the directions of the interferers. The power inversion (PI) algorithm [4,5,6] is one of the known adaptive beamforming algorithms that has been found to have abundant applications in wireless communication and navigation systems. In practice, the performance of the PI algorithm is affected by non-ideal factors of the hardware [7,8,9,10]. Typical hardware impairments include channel mismatch, I/Q imbalance, array element perturbation, etc. Studying the impacts of these factors contributes greatly to practical implementations, e.g., for proposing specific requirements on the circuit design parameters, or analyzing the bottleneck factors of the system.

At present, several non-ideal factors affecting the anti-jamming performance of the PI algorithm have been analyzed [7,8,9,10]. In [7], the effects of channel mismatch, I/Q imbalance, and antenna mutual coupling are studied. It is shown that the interference cancellation ratio (ICR) is mainly related to the I/Q imbalance and amplitude-phase perturbations of the channel, but is almost independent of antenna mutual coupling. In [8], the effects of channel uniformity are analyzed. The simulation results show that the PI algorithm can combat channel uniformity when the channel amplitude disturbance does not exceed 7 dB. In [9], the effects of array element position perturbations on the PI algorithm in navigation systems are analyzed. The results show that element position perturbations result in positioning deviations. In [10], the effects of antenna phase center drift are analyzed. The results show that the change in phase center will produce nulls in the non-interference directions, which leads to attenuation of the useful signals and, thus, degrades the signal-to-interference ratio of the output signal. However, to the best of the authors’ knowledge, the impacts of phase noise have not been studied yet, according to the open literature.

In wireless communication systems, the radio-frequency (RF) signals received by the antenna array need to be down-converted into base-band signals, in order to apply the digital PI algorithm [11,12]. The phase noise introduced in the down-conversion process may affect the system performance. This is because the phase noise has a phase modulation effect on the received signals [13], which reduces their correlation. As a result, the multi-antenna signals cannot be added coherently, such that the algorithm performance is degraded. The analysis of this effect includes aspects such as the data rates of MIMO communication and the ICR of self-interference cancellation in full-duplex systems [11,12,13,14,15,16,17,18,19,20,21,22,23]. For example, the transmitted and received phase noises degrade the EVM (error vector magnitude) of the MIMO-OFDM system, as reported in [15,16]; the received phase noise deteriorates the spectral efficiency of mmWave multi-user massive MIMO up-link transmission [11], the network capacity of cell-free massive MIMO [17], and the accuracy of the channel and location estimation of massive MIMO [18]. In addition, the decorrelation effect of phase noise reduces the self-interference cancellation performance in full-duplex systems, as reported in [12,19,20], and its joint effects with circuit nonlinearity, carrier frequency offset, and I/Q imbalance are studied in [21,22], respectively. As in these researches, there is also motivation to study the impacts of phase noise on the performance of the PI algorithm in adaptive beamforming anti-jamming systems.

In adaptive beamforming systems, there are two main types of multi-channel down-conversion circuit structures, i.e., common local oscillator and common reference clock [24,25,26,27,28], which are illustrated in Figure 1, respectively. In the first type, a single LO signal, which is generated by a phase lock loop (PLL), is fed to all the mixers. In the second type, a single reference clock is fed to multiple PLLs that generate multiple LO signals, which are then fed to multiple mixers. For the first type, the phase noises are the same for all the channels. Therefore, the correlations between the array base-band signals are not affected, so a high ICR can still be achieved. However, for the second type, the phase noises are uncorrelated, due to the independent PLLs [13]. For this reason, the correlations between the signals are deteriorated, which may degrade the algorithm performance. The reference clock has a low frequency, e.g., 10 MHz; thus, it is easier to distribute in circuits than the high-frequency LO signal, e.g., several GHz. Therefore, the common reference clock method is widely used in practice, especially for large antenna arrays or distributed antenna systems [26,28].

The aim of this paper is to establish a mathematical model to quantify the impact of phase noise on the performance of the PI algorithm, when a common reference clock-based multi-channel down-conversion circuit is employed. The main contributions are listed below:

(1) We establish a new mathematical model of the PI algorithm by including the phase noises of the common reference clock-based down-conversion circuits;

(2) We obtain a closed-form formula of ICR, which reveals the relation between the anti-jamming performance and the phase noise level, as well as the number of auxiliary antennas;

(3) We verify the theoretical results with both simulated and measured phase noise data.

The rest of the paper is organized as follows: Section 2 establishes the new mathematical model of the PI algorithm, considering phase noise. Section 3 analyzes the anti-jamming performance of the PI algorithm. Section 4 presents the numerical results. The results are discussed in Section 5. Conclusions are drawn in Section 6.

## 2. System Model

The block diagram of the digital beamforming anti-jamming system based on the PI algorithm is shown in Figure 2. The antenna array is composed of N + 1 (N is the number of auxiliary antennas) identical antenna elements, of which the first one is taken as the main antenna and the rest are auxiliary antennas. For simplicity, we assume that all the antenna elements are isotropic antennas. The signals received by the antenna array pass through the LNAs, down-conversion blocks, and ADCs, and are then processed by the PI algorithm before they are finally output to the receiver. The N + 1 down-conversion blocks use different LO signals generated by N + 1 independent PLLs with a common reference clock.

Denote the signal received by the main antenna as $x_{0}(t)=a_{0}(\theta )\left(s(t)+u(t)\right)$, and the signals received by the auxiliary antennas as x(t), which can be written as follows:(1)$$\begin{array}{cc} x(t) & ={\left[x_{1}(t),x_{2}(t),...,x_{N}(t)\right]}^{T}\\ & ={\left[a_{1}(\theta ),a_{2}(\theta ),\ldots ,a_{N}(\theta )\right]}^{T}\left(s(t)+u(t)\right)\\ & =a(\theta )\left(s(t)+u(t)\right) \end{array}$$
where $a_{i}(\theta )$, i=0,…,N is the response of the *i*-th antenna element and a(θ) is the response vector of the antenna array.

The received base-band signals of the main antenna and the auxiliary array are as follows:(2)$$x_{0}(n)=a_{0}(\theta )\left(s(n)+u(n)\right)e^{j\varphi_{0}(n)+j\phi_{0}}$$
(3)$$x_{i}(n)=a_{i}(\theta )\left(s(n)+u(n)\right)e^{j\varphi_{i}(n)+j\phi_{i}}\begin{array}{cc} , & \end{array}i=1,2,...,N$$
where $\phi_{i}$ is the phase shift due to the circuit of the *i*-th channel; $\varphi_{i}(n)$ is the phase noise of the *i*-th LO signal; s(n) is the interference signal; u(n) is the useful signal. In order to focus on the impacts of phase noise, we ignore the thermal noise of the receiver.

## 3. Anti-Jamming Performance Analysis

After being processed by the PI algorithm, the output signal e(n) can be expressed as follows:(4)$$e(n)=x_{0}(n)-w^{H}x(n)$$
where $w={\left[w_{1},w_{2},\ldots ,w_{N}\right]}^{T}$ is the beamforming weight vector. The optimal weight based on the PI algorithm is obtained by the following equation [4]:(5)$$w_{opt}=R^{-1}p$$
where $R=E\left\{x(n)x{\left(n\right)}^{H}\right\}$ is the autocorrelation matrix of the signals received by the auxiliary antennas, and $p=E\left\{x(n)x_{0}{}^{*}\left(n\right)\right\}$ is the cross-correlation vector of the signals received by the main antenna and the auxiliary antennas.

Substituting the optimum weight $w_{opt}$ into Equation (4), we can obtain the minimum power of e(n) as follows:(6)$$P_{e}=E\left\{{\left|e\left(n\right)\right|}^{2}\right\}=P_{0}-p^{H}R^{-1}p$$
where $P_{0}=E\left\{{\left|x_{0}(n)\right|}^{2}\right\}$ is the power of the signals received by the main antenna. It can be observed that the minimum power of the output signals is determined by R and p, which are closely related to the number of auxiliary antennas and the phase noises of the LO signals of the down-conversion circuit.

Suppose that the useful signal is independent from the interference signal, R can be expressed as follows:(7)$$R=R_{ss}+R_{uu}$$
where $R_{ss}$ and $R_{uu}$ represent the autocorrelation matrix of the interference signals and the useful signals, respectively. The PI algorithm is mainly applicable to situations with strong interference signals, but weak useful signals, such that $R\approx R_{ss}$ [29].

The element of row *i* and column *j* of matrix R can be obtained as follows:(8)$$R_{ij}=a_{i}(\theta )a_{j}{}^{*}(\theta )E\left\{e^{j\left[\varphi_{i}(n)-\varphi_{j}(n)+\phi_{i}-\phi_{j}\right]}\right\}P_{s}$$

For i=j, we can obtain the following:(9)$$R_{ii}=\left|a_{i}{}^{2}(\theta )\right|P_{s}$$

For i≠j, since the phase noises of different down-conversion blocks are independent, we can obtain the following:(10)$$R_{ij}=a_{i}(\theta )a_{j}{}^{*}(\theta )e^{j(\phi_{i}-\phi_{j})}E\left\{e^{j\varphi_{i}(n)}\right\}E\left\{e^{-j\varphi_{j}(n)}\right\}P_{s}$$

The *k*-th element of vector p is as follows:(11)$$p_{k}=a_{k}(\theta )a_{0}{}^{*}(\theta )E\left\{e^{j\left[\varphi_{k}(n)-\varphi_{0}(n)+\phi_{k}-\phi_{0}\right]}\right\}P_{s}$$
where $P_{s}$ is the power of the interference signal s(n).

Similarly, $p_{k}$ can be expressed as follows:(12)$$p_{k}=a_{k}(\theta )a_{0}{}^{*}(\theta )e^{j(\phi_{k}-\phi_{0})}E\left\{e^{j\varphi_{k}(n)}\right\}E\left\{e^{-j\varphi_{0}(n)}\right\}P_{s}$$

The phase noise of a PLL conforms to the stationary Gaussian colored noise model [13]. Therefore, the probability density function of the phase noise at any instant is as follows:(13)$$P(\varphi )=\frac{1}{\sqrt{2\pi \sigma^{2}}}e^{-\frac{\varphi^{2}}{2\sigma^{2}}}$$
where $\sigma^{2}$ is the variance.

We can then find the following:(14)$$\begin{array}{cc} E\left\{e^{j\varphi }\right\} & =\int_{-\infty }^{+\infty }P(\varphi )\cdot e^{j\varphi }d\varphi =\frac{1}{\sqrt{2\pi \sigma^{2}}}\int_{-\infty }^{+\infty }e^{-\frac{{\left(\varphi -j\sigma^{2}\right)}^{2}}{2\sigma^{2}}-\frac{\sigma^{2}}{2}}d\varphi \\ & =e^{-\frac{\sigma^{2}}{2}}\cdot \int_{-\infty }^{+\infty }\frac{1}{\sqrt{2\pi \sigma^{2}}}\cdot e^{-\frac{{\left(\varphi -j\sigma^{2}\right)}^{2}}{2\sigma^{2}}}d\varphi =e^{-\frac{\sigma^{2}}{2}} \end{array}$$

Similarly, $E\left\{e^{-j\varphi }\right\}=e^{-\frac{\sigma^{2}}{2}}$. Therefore, we can obtain the following:(15)$$R_{ij}=a_{i}(\theta )a_{j}{}^{*}(\theta )e^{j(\phi_{i}-\phi_{j})}e^{-\frac{\sigma_{i}{}^{2}+\sigma_{j}{}^{2}}{2}}P_{s},\begin{array}{cc} & \left(i\neq j\right) \end{array}$$
(16)$$p_{k}=a_{k}(\theta )a_{0}{}^{*}(\theta )e^{j(\phi_{k}-\phi_{0})}e^{-\frac{\sigma_{k}{}^{2}+\sigma_{0}{}^{2}}{2}}P_{s}$$

It can be observed from Equations (15) and (16) that R and p are related to the phase noise variance $\sigma^{2}$. According to [30], $\sigma^{2}$ can be obtained as follows:(17)$$\sigma^{2}=2\int_{0}^{B}\frac{S_{\varphi }(f)}{C}df$$
where $S_{\varphi }(f)$ is the single side-band power spectral density of the phase noise; C is the carrier power; B is the bandwidth of the phase noise.

In order to simplify the analysis and focus on the impact of phase noise, but without affecting the conclusions of the paper, we make the following assumptions:

(1) The antenna array is a linear or planar array with isotropic elements; the angle of the interference signal is 0 degrees; thus, the array response is $a(\theta )=1_{N}$.

(2) The fixed phase shifts of all the channels are equal, i.e., $\phi_{0}=\phi_{1}=\ldots =\phi_{N}$.

According to the above assumptions, $R_{ij}$ can be simplified into the following:(18)$$R_{ij}=P_{s}\cdot e^{-\frac{\sigma_{i}{}^{2}+\sigma_{j}{}^{2}}{2}}$$

$p_{k}$ can be simplified into the following:(19)$$p_{k}=P_{s}\cdot e^{-\frac{\sigma_{k}{}^{2}+\sigma_{0}{}^{2}}{2}}$$

From Equations (18) and (19), it can be observed that each element of matrix R or vector p is only related to the power of the interference signal and the variances of the phase noises.

Suppose that the LO signals have the same phase noise level, then the variance of the phase noises of the down-conversion blocks is equal, i.e., $\sigma_{0}{}^{2}=\sigma_{1}{}^{2}=\ldots =\sigma_{N}{}^{2}=\delta^{2}$.

Then, the matrix R can be written as follows:(20)$$R=P_{s}\left[\begin{array}{cccc} 1 & e^{-\delta^{2}} & \cdots & e^{-\delta^{2}}\\ e^{-\delta^{2}} & 1 & \cdots & e^{-\delta^{2}}\\ \vdots & \vdots & \cdots & \vdots \\ e^{-\delta^{2}} & e^{-\delta^{2}} & \cdots & 1 \end{array}\right]$$

The vector p can be written as follows:(21)$$p=P_{s}\left[\begin{array}{c} e^{-\delta^{2}}\\ e^{-\delta^{2}}\\ \vdots \\ e^{-\delta^{2}} \end{array}\right]$$

Then, we turn to find the inverse matrix of R (see Appendix A for the detailed derivation process), and obtain the following:(22)$$R^{-1}=\frac{1}{P_{s}}\left[\begin{array}{cccc} x & y & \cdots & y\\ y & x & \cdots & y\\ \vdots & \vdots & \cdots & \vdots \\ y & y & \cdots & x \end{array}\right]$$
where,
(23)$$\left\{ \begin{array}{l} x=\frac{2e^{-\delta^{2}}-e^{-\delta^{2}}N-1}{e^{-2\delta^{2}}(N-1)-e^{-\delta^{2}}(N-2)-1}\\ y=\frac{e^{-\delta^{2}}}{e^{-2\delta^{2}}(N-1)-e^{-\delta^{2}}(N-2)-1} \end{array}\right.$$

Then, the minimum power of the output signal can be derived as follows:(24)$$P_{e}=P_{I}-p^{H}R^{-1}p=P_{s}\frac{\left(e^{-\delta^{2}}-1)(Ne^{-\delta^{2}}+1\right)}{e^{-\delta^{2}}-Ne^{-\delta^{2}}-1}$$

In this paper, we use ICR to measure the anti-jamming performance of the PI algorithm, which is defined as the ratio of the received signal power of the main antenna to the beamformer output signal [7], i.e., the following:(25)$$ICR=\frac{P_{I}}{P_{e}}=\frac{e^{-\delta^{2}}-Ne^{-\delta^{2}}-1}{\left(e^{-\delta^{2}}-1)(Ne^{-\delta^{2}}+1\right)}$$

From the above formula, it can be observed that ICR is determined by the variance of the phase noises and the number of auxiliary antennas. Special cases are N=1 and N→∞, for which we obtain the following:

(1) For N=1,
(26)$$ICR_{N=1}=\frac{1}{1-e^{-2\delta^{2}}}$$

(2) For N→∞,
(27)$$ICR_{N\to \infty }=\frac{1}{1-e^{-\delta^{2}}}$$

Therefore, as N increases from one to infinity, the increment in ICR (in decibel) is upper-bounded to be the following:(28)$$\Delta ICR=10\log_{10}\left(1+e^{-\delta^{2}}\right)$$

Since $\delta^{2}$ is small, we can further obtain that the maximum ΔICR is around 3 dB.

## 4. Numerical Results

In this section, we present the simulation results to verify the theoretical derivations obtained in Section 3. The analysis consists of the following three parts: (1) ICR versus phase noise level for a given number of auxiliary antennas; (2) ICR versus the number of auxiliary antennas for a given phase noise level; (3) verification based on the measured phase noise data.

The simulation model is illustrated in Figure 3. A single-tone interference signal, with a frequency of 100 Hz and a sampling rate of 2 MHz, first passes through a linear antenna array; the array output is then fed into multiple phase noise blocks to capture the effect of the multi-channel down-conversion circuit. After this, the output signals are processed by the PI algorithm block to assess the ICR performance. The phase noise model in MATLAB Communications toolbox is used therein, which takes the phase noise levels at different frequency offsets as the input [31].

We use the widely used phase noise model in [32] to simulate the power spectrum of the phase noise, which can be expressed as follows:(29)$$S_{\varphi }(f)=10^{-c}+\left\{ \begin{array}{l} 10^{-a},\left|f\right|\leq f_{l}\\ 10^{-(f-f_{l})\frac{b}{f_{h}-f_{l}}-a},f_{l}<f<f_{h}\\ 10^{(f+f_{l})\frac{b}{f_{h}-f_{l}}-a},-f_{h}<f<-f_{l} \end{array}\right.$$
where $f_{l}$ determines when the phase noise level begins to decline; $f_{h}$ is where the noise floor becomes dominant; a determines the phase noise level near the carrier; b determines the steepness of the linear slope of the curve; c determines the noise floor.

In the following simulation, we use the above model for analysis and set the parameters as the following: $f_{l}=1\;\mathrm{kHz}$, $f_{h}=1\;\mathrm{MHz}$, b=100, and c=11. We vary the parameter a to change the phase noise level near the carrier ($0<f\leq f_{l}$). For example, when a=6, the phase noise level is −60 dBc/Hz@1 kHz, and the power spectrum is shown in Figure 4.

### 4.1. ICR versus Phase Noise Level

In this section, we analyze the ICR performance of the PI algorithm versus the phase noise level for different auxiliary array sizes.

Since the theoretical ICR performance is related to the phase noise variance $\delta^{2}$ (in unit of rad^2^), we first analyze the phase noise variance using Equation (17). The result is presented in Figure 5.

The phase noise level varies from −90 dBc/Hz to −50 dBc/Hz at 1 kHz frequency offset, which covers the typical phase noise level in practice. We can observe that when the phase noise level is between −80 dBc/Hz@1 kHz and −50 dBc/Hz@1 kHz, the phase noise variance increases logarithmically linearly with the phase noise level; when the phase noise level is less than −80 dBc/Hz@1 kHz, which is close to the noise floor, the relation between the two parameters no longer remains logarithmically linear.

Figure 6 shows the ICR versus phase noise variance when the auxiliary antenna number N equals 1, 10 and 100. As we can observe, the ICR (in decibel) decreases logarithmically linearly with the phase noise variance. Specifically, when the phase noise variance increases 10 times, the ICR decreases by about 10 dB. The theoretical results are consistent with the simulation results. However, due to the approximations made in the mathematical derivations, the ICR equation slightly underestimates the performance by about 1 dB, as compared to the simulation results. This underestimation is consistent for all auxiliary array sizes.

### 4.2. ICR versus Number of Auxiliary Antennas

In this section, we analyze the ICR performance versus the number of auxiliary antennas for certain phase noise levels.

The considered phase noise levels are −80 dBc/Hz@1 kHz, −65 dBc/Hz@1 kHz, and −50 dBc/Hz@1 kHz. The corresponding phase noise variances are 1.27 × 10^−4^, 3.40 × 10^−3^, and 1.07 × 10^−1^, respectively. The number of auxiliary antennas increases from 1 to 100. The other simulation parameters are kept the same as those in the last subsection. The results are shown in Figure 7.

It can be observed that the theoretical results are consistent with the simulation results. For a given phase noise level, the ICR increases with the number of auxiliary antennas, but levels off at around N=10. For the three cases herein, the maximum increments are 3.47 dB, 3.27 dB, and 3.25 dB, respectively. This is the same as the theoretical analysis, as given in Equation (28), i.e., the ICR increment is upper-bounded to 3 dB.

### 4.3. Analysis Based on Measured Phase Noise Data

In this section, we use the measured phase noise data to evaluate the ICR performance.

Three types of down-conversion modules are considered, which are named type A, B, and C, respectively. Type A is the Norsat 1008XHB Ku-Band LNB; type B is the NJR2936E Ku-Band LNB; type C is our customized module (designed to improve the phase noise performance). The phase noise spectrums are measured using a spectrum analyzer, and then they are fed into the simulation model used in the former subsections to analyze the ICRs. The measured power spectrums are shown in Figure 8, and the corresponding phase noise variances are calculated using Equation (17). Due to the loss of low frequency [33], the spectrum analyzer cannot accurately measure the phase noise level near the carrier. Therefore, we use the phase noise level at 100 Hz frequency offset in place of the real values of 0~100 Hz frequency offsets. The computed phase noise variances are shown in Table 1.

Let the auxiliary array size be *N* = 1, 10, and 100, the theoretical and simulated ICRs using three types of down-conversion blocks are shown in Table 2. Figure 9 presents the results of ICR versus auxiliary array size.

From these results, we can observe the following:

(1) Different performances are achieved for the three types of down-conversion blocks. However, type C provides the best performance, due to its lower phase noise (see Figure 8), offering an improvement of around 3 dB;

(2) Using the measured phase noise data, the theoretical results are consistent with the simulated results, which is the same as the above subsections, using the simulated phase noise data. Although our theory tends to underestimate the performance, the underestimation errors are marginal. A particular case is the type B block, for which the performance is underestimated by around 2 dB, greater than the other two types. This is mainly due to the spurious signals of the 100~1000 Hz frequency offsets (see Figure 8), which may violate the phase noise model.

## 5. Discussion

In practice, the phase noise power spectrum can be easily measured. Thus, the proposed mathematical model can accurately predict the ICR performance using the measured data. Conversely, given a target ICR, the phase noise variance requirement can also be calculated directly.

Since there is a straightforward relation between the phase noise variance and phase noise power spectrum (Equation (17)), the circuit designer can first simulate the phase noise power spectrum, then convert it to phase noise variance and compare it with the required value to validate the circuit design. For example, to achieve a typical ICR of 40 dB, by using three auxiliary antennas, the required phase noise variance is 7.5 × 10^−5^ rad^2^.

According to traditional models, increasing the number of auxiliary antennas can increase the array gain, and, therefore, can improve the ICR performance. However, according to our theory and simulation results (see Equation (28) and Table 2), the ICR increases by, at most, 3 dB, regardless of the phase noise level. From the simulation results in Figure 8 and Figure 9, the ICR closely approaches the upper bound when the number of auxiliary antennas increases to 10. Therefore, if phase noise is the bottleneck factor, it is of little help to further increase the number of auxiliary antennas.

## 6. Conclusions

In this paper, the impacts of phase noise on the anti-jamming performance of the PI algorithm in an adaptive beamforming system are analyzed, which employs a common reference clock-based down-conversion circuit. First, by introducing the phase noises of the local oscillator signals of the down-conversion circuit, a new mathematical model of the PI algorithm is established. Second, a closed-form formula of ICR is derived, which shows the quantitative relation between the anti-jamming performance and the phase noise level, as well as the number of auxiliary antennas. Third, the theoretical analysis is verified by the simulations using simulated and measured phase noise data. The proposed mathematical model slightly underestimates the ICR performance by about 1 dB, in comparison with the simulation results. The main conclusions are as follows:

(1) The ICR in decibel decreases logarithmically linearly with the phase noise variance of the local oscillator, which is applicable to all array sizes. In other words, when the phase noise variance increases ten times, the ICR decreases by 10 dB.

(2) For a given phase noise level, the ICR increases with the number of auxiliary antennas, but the increment is upper-bounded by around 3 dB.

## Figures and Tables

**Figure 1 sensors-22-02362-f001:**
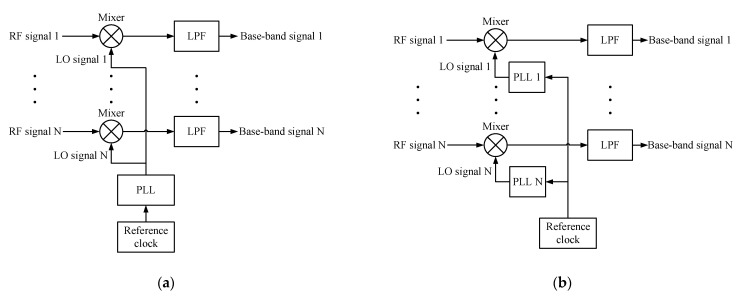
Typical down-conversion circuit structures: (**a**) with a common local oscillator; (**b**) with a common reference clock.

**Figure 2 sensors-22-02362-f002:**
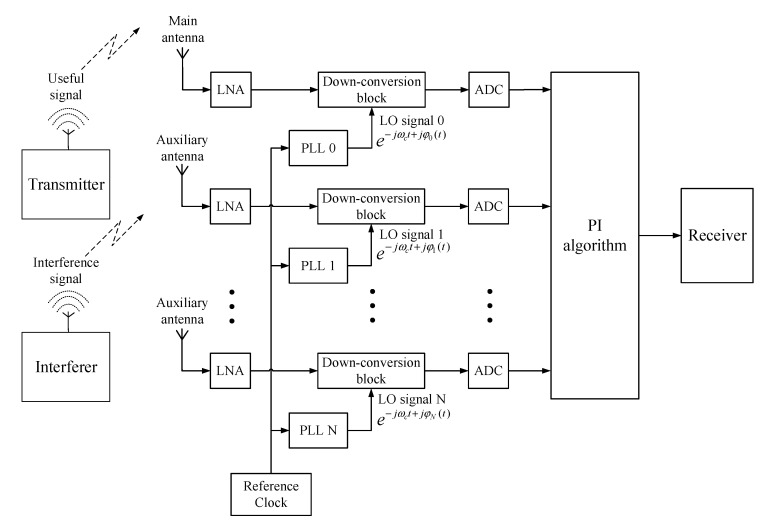
Block diagram of digital beamforming anti-jamming system based on the PI algorithm.

**Figure 3 sensors-22-02362-f003:**
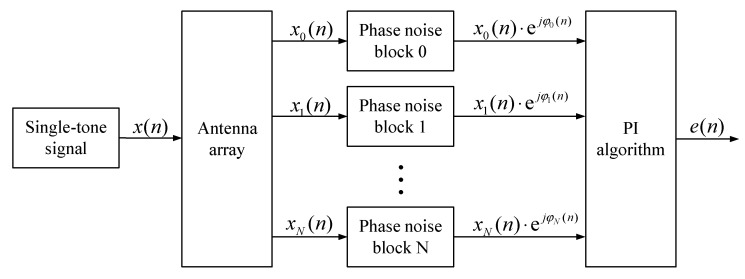
Simulation method.

**Figure 4 sensors-22-02362-f004:**
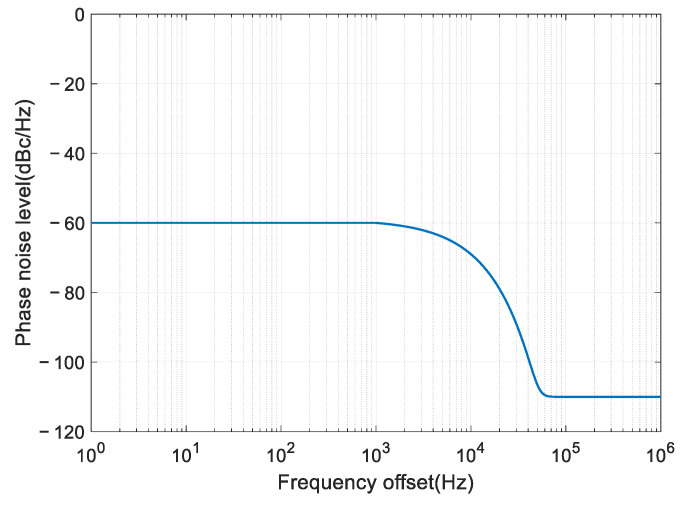
The power spectrum of phase noise when a=6.

**Figure 5 sensors-22-02362-f005:**
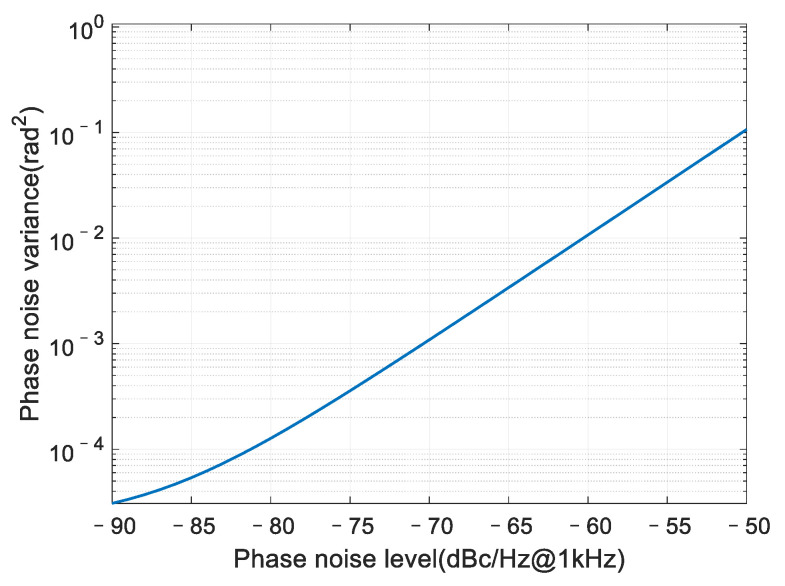
Relation between the variance and power spectrum of phase noise.

**Figure 6 sensors-22-02362-f006:**
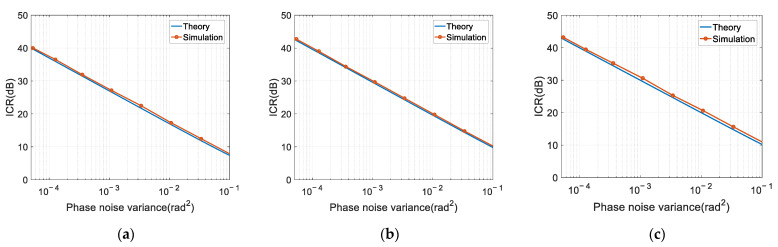
ICR versus phase noise variance: (**a**) *N* = 1; (**b**) *N* = 10; (**c**) *N* = 100.

**Figure 7 sensors-22-02362-f007:**
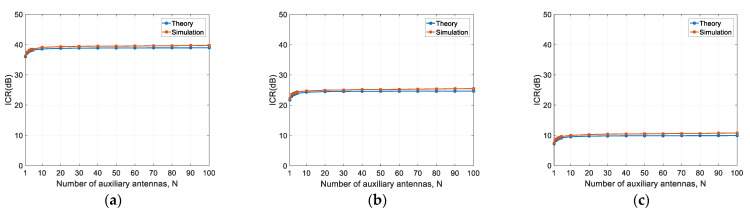
ICR versus the number of auxiliary antennas: (**a**) $\delta^{2}=1.27\times 10^{-4}\;{\mathrm{rad}}^{2}$; (**b**) $\delta^{2}=3.40\times 10^{-3}\;{\mathrm{rad}}^{2}$; (**c**) $\delta^{2}=1.07\times 10^{-1}\;{\mathrm{rad}}^{2}$.

**Figure 8 sensors-22-02362-f008:**
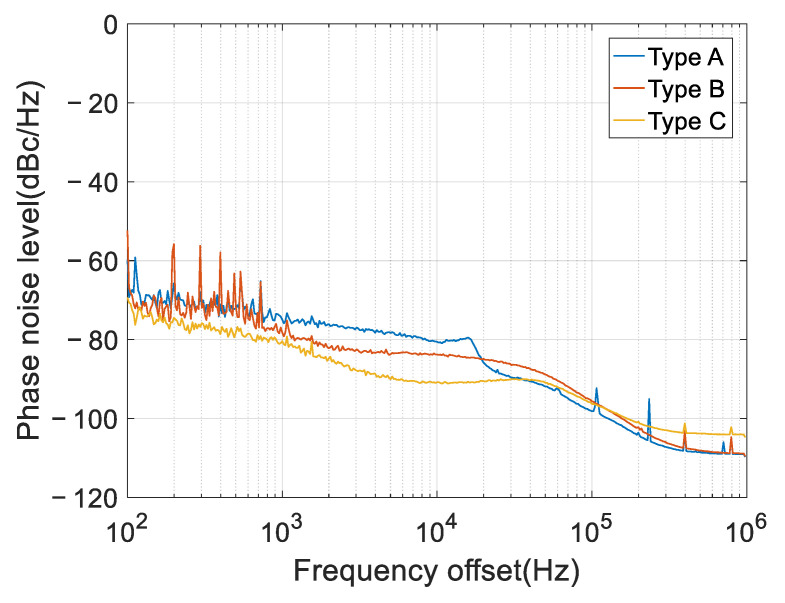
The measured phase noise spectrum.

**Figure 9 sensors-22-02362-f009:**
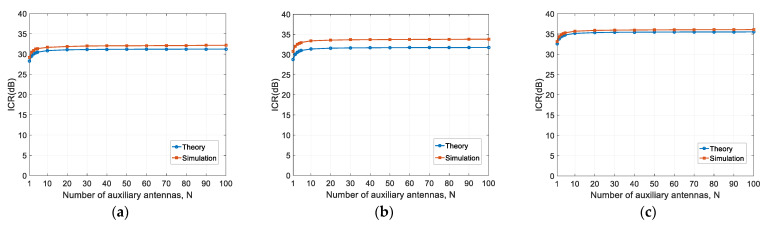
ICR versus the auxiliary array size: (**a**) type A; (**b**) type B; (**c**) type C.

**Table 1 sensors-22-02362-t001:** The phase noise variance of the three types of down-conversion blocks.

Down-Conversion Block	Type A	Type B	Type C
Phase noise variance (rad^2^)	7.46 × 10^−4^	6.62 × 10^−4^	2.77 × 10^−4^

**Table 2 sensors-22-02362-t002:** The values of ICR.

Auxiliary Array Size	ICR	Type A	Type B	Type C
*N* = 1	Theory	28.3	28.8	32.6
	Simulation	29.2	30.8	33.1
*N* = 10	Theory	30.9	31.4	35.2
	Simulation	31.7	33.3	35.7
*N* = 100	Theory	31.2	31.8	35.5
	Simulation	32.2	33.8	36.1

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
