# Peer review of "Impacts of Phase Noise on the Anti-Jamming Performance of Power Inversion Algorithm"

_sensors, 2022, doi:10.3390/s22062362_

Round 1

Reviewer 1 Report

Review of the article „ Impacts of Phase Noise on the Anti-jamming Performance of Power Inversion Algorithm“ by authors: Minglei Zhou, Qing Wang, Fangmin He, and Jin Meng

Shortcomings of the article:

In article show old literature sources. About 65 percent of cited articles more 5 years old. In the journal should be overviewed the only new information. The literature review needs to be extended to include new references. In addition, the literature review is very modest and needs to be expanded.

The aim of the research needs to be clearly stated and presented at the end of the introduction.

The article lacks an analysis of the results therefore needs add discussions sections.

What is the reliability of the results obtained?

Not possible to end a chapter with a table or a picture (Chapter 4).

In the conclusions must clearly show what problems the researchers have solved and how much to get results are better than the results of other researches. The conclusions should be clear and concise with the numerical values provided to support and justify the results obtained.

Author Response

Dear Reviewer:

We sincerely thank you for your valuable comments and suggestions, which are highly insightful and enable us to greatly improve the quality of our manuscript. According to your comments, we have revised our manuscript thoroughly. The modifications are highlighted with RED color in the new manuscript. Our point-by-point replies to each of your comments are listed below.

Point 1: In article show old literature sources. About 65 percent of cited articles more 5 years old. In the journal should be overviewed the only new information. The literature review needs to be extended to include new references. In addition, the literature review is very modest and needs to be expanded.

Response 1: According to your suggestion, we extend the literature review, especially the papers published in recent five years. Specifically, we have added literature about the latest applications of adaptive beamforming and PI algorithm and the latest research on the impacts of phase noise in MIMO communication and self-interference cancellation, which provided valuable references for our study. After the improvement, the references in recent five years have reached 70 percent.

Point 2: The aim of the research needs to be clearly stated and presented at the end of the introduction.

Response 2: According to your suggestion, we have added the aim of our research at the end of the introduction. That is, “The aim of this paper is to establish a mathematical model to quantify the impact of phase noise on the performance of PI algorithm, when a common reference clock based multi-channel down-conversion circuit is employed.”

Point 3: The article lacks an analysis of the results therefore needs add discussions sections.

Response 3: According to your suggestion, we have added Section 5 for discussion in the revised manuscript. In the discussion section, we analyze the results of this paper and discuss how the proposed theoretical models and simulation results can be used for parameter selection and system design.

The added section is copied here:

5.Discussion

In practice, the phase noise power spectrum can be easily measured. Thus, the proposed mathematical model can accurately predict the ICR performance using the measured data. Or, conversely, given a target ICR, the phase noise variance requirement can also be calculated directly.

Since there is a straightforward relation between the phase noise variance and phase noise power spectrum (Equation (17)), the circuit designer can first simulate the phase noise power spectrum, then convert to phase noise variance and compare with the required value to validate the circuit design. For example, to achieve a typical ICR of 40dB, by using 3 auxiliary antennas, the required phase noise variance is 7.5×10-5rad2.

According to traditional models, increasing the number of auxiliary antennas can increase the array gain and therefore can improve the ICR performance. However, according to our theory and simulation results (see Equation (28) and Table (2)), ICR increases by at most 3dB regardless of phase noise level. From the simulation results in Figure 8 and 9, ICR closely approaches the upper bound, when the number of auxiliary antennas grows to 10. Therefore, if phase noise is the bottleneck factor, it is of little help to further increase the number of auxiliary antennas.

Point 4: What is the reliability of the results obtained?

Response 4: In this paper, the simulation results are obtained using both simulated and measured phase noise data. It can be found that all simulation results are consistent with the theoretical model. In addition, the numerical results are obtained by extensive Monte Carlo simulations, so as to guarantee the reliability of the results.

Point 5: Not possible to end a chapter with a table or a picture (Chapter 4).

Response 5: According to your suggestion, we have optimized the layout of the paper to avoid the typesetting problems.

Point 6: In the conclusions must clearly show what problems the researchers have solved and how much to get results are better than the results of other researches. The conclusions should be clear and concise with the numerical values provided to support and justify the results obtained.

Response 6: According to your suggestion, we have revised the presentation of the conclusion to state our research results more clearly. According to the literature, the impact of phase noise on PI algorithm performance has not been studied yet. Our work adds new knowledge to the field regarding the effects of hardware non-idealities. In addition, numerical values are provided in the conclusion to support our theoretical model. In particular, the simulations show that the theoretical model underestimates the ICR performance by around 1dB, which justifies the accuracy our model.

The modified conclusion is:

In this paper, the impacts of phase noise on the anti-jamming performance of PI algorithm in adaptive beamforming system are analyzed, which employs a common reference clock based down-conversion circuits. First, by introducing the phase noises of local oscillator signals of down-conversion circuits, a new mathematical model of PI algorithm is established. Second, a closed-form formula of ICR is derived, which shows the quantitative relation of anti-jamming performance with respect to phase noise level and the number of auxiliary antennas. Third, the theoretical analysis is verified by the simulations using simulated and measured phase noise data. The proposed mathematical model slightly underestimates the ICR performance by about 1 dB in comparison with the simulation results. The main conclusions are the following:

1) The ICR in decibel decreases logarithmically linearly with the phase noise variance of the local oscillator, which is applicable to all array sizes. In other words, when phase noise variance increases ten times, ICR decreases by 10dB.

2) For a given phase noise level, the ICR increases with the number of auxiliary antennas, but the increment is upper-bounded by around 3dB.

Reviewer 2 Report

Dear Authors,

The paper is well organised and deals with an interesting topic. The only concern that the reviewer has is that the cited references are old. In other words, there are very few references of recently published paper. This fact suggests that the author is not aware of the most recent advances in the field or that there has been little progress in the field. To overcome this issue, the reviewer suggests to find several recently published papers to reference.  

Author Response

Dear Reviewer:

We sincerely thank you for your valuable comments and suggestions, which are highly insightful and enable us to greatly improve the quality of our manuscript. According to your comments, we have revised our manuscript thoroughly. The modifications are highlighted with RED color in the new manuscript. Our point-by-point replies to each of your comments are listed below.

Point 1: The only concern that the reviewer has is that the cited references are old. In other words, there are very few references of recently published paper. This fact suggests that the author is not aware of the most recent advances in the field or that there has been little progress in the field. To overcome this issue, the reviewer suggests to find several recently published papers to reference.

Response 1: According to your suggestion, we extend the literature review, especially the papers published in recent five years. Specifically, we have added literature about the latest applications of adaptive beamforming and PI algorithm and the latest research on the impacts of phase noise in MIMO communication and self-interference cancellation, which provided valuable references for our study. After the improvement, the references in recent five years have reached 70 percent.

Reviewer 3 Report

The paper deals with performance assessment of an interference-nulling antenna array system, where each antenna element has a local oscillator locked to a common reference clock. It is an interesting scenario, as this solution is preferable for its lower cost with respect to a system employing LO carrier distribution.

The authors study the impact of uncorrelated phase noise errors between antenna elements on the interference canceling performance, in terms of Interference Cancelling Ratio (ICR).

The results show a good agreement between the reported approximate theoretical analysis and simulations, and some performance results are shown for three types of down-conversion modules, one of which was designed by the authors but not detailed in the paper.

The presented results show that the simulation results are in good agreement with the approximate formula derived by the authors, so that antenna array specifications, in terms of phase noise variance and number of elements may be defined without resorting to simulations.

Some English phrasing could be improved, at least from this reviewer point of view, but the paper is overall readable in its current form.

Author Response

Dear Reviewer:

We sincerely thank you for your valuable comments and suggestions, which are highly insightful and enable us to greatly improve the quality of our manuscript. According to your comments, we have revised our manuscript thoroughly. The modifications are highlighted with RED color in the new manuscript. Our point-by-point replies to each of your comments are listed below.

Point 1: Some English phrasing could be improved, at least from this reviewer point of view, but the paper is overall readable in its current form.

Response 1: We agree that the writing should be improved for better readability. According to your suggestions, the new manuscript is reviewed by our college, Zhong Liu and Kai Yang, who have published many English papers in MDPI and IEEE journals. We believe the new manuscript is better presented than the last one. We will further improve the paper with the help of proofreading services.

Round 2

Reviewer 1 Report

Not possible to end a chapter with a picture (Chapter 4, page 8 and page 11).

Author Response

Dear Reviewer:

We sincerely thank you for your valuable comments and suggestions, which are highly insightful and enable us to greatly improve the quality of our manuscript. According to your comments, we have revised our manuscript thoroughly. The modifications are highlighted with RED color in the new manuscript. Our point-by-point replies to each of your comments are listed below.

Point 1: Not possible to end a chapter with a picture (Chapter 4, page 8 and page 11).

Response 1: In the new manuscript, we place the figures between the texts or at the top of the page. This avoids ending a chapter with a picture.
